# Antibacterial Activity of Some Molecules Added to Rabbit Semen Extender as Alternative to Antibiotics

**DOI:** 10.3390/ani11041178

**Published:** 2021-04-20

**Authors:** María Pilar Viudes-de-Castro, Francisco Marco-Jimenez, José S. Vicente, Clara Marin

**Affiliations:** 1Centro de Investigación y Tecnología Animal, Instituto Valenciano de Investigaciones Agrarias (CITA-IVIA), Polígono La Esperanza n° 100, 12400 Segorbe, Spain; 2Instituto de Ciencia y Tecnología Animal, Universitat Politècnica de València, 46022 Valencia, Spain; fmarco@dca.upv.es (F.M.-J.); jvicent@dca.upv.es (J.S.V.); 3Departamento de Producción y Sanidad Animal, Salud Pública Veterinaria y Ciencia y Tecnología de los Alimentos, Instituto de Ciencias Biomédicas, Facultad de Veterinaria, Universidad Cardenal Herrera-CEU, CEU Universities, Avenida Seminario s/n, 46113 Moncada, Spain; clara.marin@uchceu.es

**Keywords:** artificial insemination, antibiotics, antibacterial activity, semen quality, reproductive performance

## Abstract

**Simple Summary:**

This study was conducted to evaluate the antibacterial activity of two aminopeptidase inhibitors and chitosan-based nanoparticles in liquid-stored rabbit semen. This study reports that the aminopeptidase inhibitors used to prevent bacterial growth could be used in semen extender as a suitable alternative to antibiotics.

**Abstract:**

Although great attention is paid to hygiene during semen collection and processing, bacteria are commonly found in the semen of healthy fertile males of different species. As the storage of extended semen might facilitate bacterial growth, extenders are commonly supplemented with antibiotics. This study aimed to evaluate the antibacterial activity of ethylenediaminetetraacetic acid (EDTA), bestatin and chitosan-based nanoparticles added to rabbit semen extender and their effect on reproductive performance under field conditions. Four different extenders were tested, supplemented with antibiotics (TCG+AB), with EDTA and bestatin (EB), with EDTA, bestatin and chitosan-based nanoparticles (QEB) or without antibiotics (TCG-AB). Extended semen was cooled at 15 °C for three days. Cooled samples were examined for bacterial growth and semen quality every 24 h for 3 days. The enterobacteria count increased considerably during storage at 72 h in semen extended with TCG+AB and TCG-AB, while extenders EB and QEB showed a bacteriostatic effect over time. After 24, 48 and 72 h, quality characteristics were retained in all groups, with no significant motility differences, either in acrosome integrity, membrane functionality or the viability of spermatozoa. Additionally, bacterial concentration present in fresh semen did not affect reproductive performance. In conclusion, EDTA and bestatin exerted a potent bacteriostatic effect over time and could be used as an alternative to conventional antibiotics in rabbit semen extenders.

## 1. Introduction

Artificial insemination (AI) is a highly efficient assisted reproductive technology used worldwide in animal breeding. Semen from healthy fertile males from different species contains bacteria stemming from natural colonisation in the male tract and the environment, despite the application of strict hygienic measures during collection and manipulation processes [1], so international directives stipulate the addition of antibiotics to semen extenders to prevent bacterial growth. However, as a consequence of excessive antibiotics use in different fields such as human medicine, veterinary medicine, livestock and fish production, agriculture and food technology, microbial resistance has emerged as one of the main concerns worldwide [2,3].

Semen contaminated with bacteria such as *Enterobacteriaceae* might adversely affect the quality of semen used for AI [4], in addition it is an important hygiene indicator [5,6,7,8]. Even though most of them are non-pathogenic bacteria, they can negatively influence sperm quality and longevity if present in high concentrations [6,7,9,10]. Additionally, several studies have shown that the seminal plasma of different species contains aminopeptidases [11,12,13,14,15,16]. Aminopeptidase activity promotes the proliferation of many bacteria that act as virulence factors, essential for the survival and maintenance of many microbial pathogens [17,18].

Nowadays, bacterial resistance to commonly used antibiotics and the global spread of resistance genes has become a serious health problem. Bacteria could counteract the actions of antimicrobials through different mechanisms, such as enzyme modification, alteration of the target binding sites, active efflux pumps or decreased permeability of bacterial membrane [19]. This resistance may occur through spontaneous mutations or by the horizontal transfer of mobile genetic elements from other bacteria, phages and/or the transmission of resistance genes from the environment [19,20]. In this sense, the transmission of resistance genes plays an important role in the spread of antimicrobial resistance among strains [19,20] and efforts need to be made to replace conventional antibiotics in the animal production industry. Some alternatives to conventional antibiotics in semen extender, such as colloid centrifugation [21,22,23,24] or removal of seminal plasma [25], have been investigated in different species. However, these techniques involve an increase in the processing time of semen. Others approaches for overcoming bacterial growth include the use of active molecules such as EDTA, [26], chitosan [27,28], nanoparticles [29], peptides [30,31] and aminopeptidase inhibitors [18,32], etc., with recognised antibacterial activity. EDTA is known to increase the outer cell wall permeability of Gram-negative bacteria, allowing other molecules easier access into the cell, facilitating an alteration or inhibition of its metabolism [33,34,35]. Chitosan, a biocompatible biodegradable and non-toxic polycationic copolymer extensively used as material for encapsulation and controlled release of chemicals [36], interacts with the bacterial cell membrane and causes cell lysis [37,38]. Furthermore, the encapsulation of GnRH in chitosan-based nanoparticles added to the extender supplemented with EDTA and bestatin can overcome the poor stability of the GnRH analogues in the presence of aminopeptidases [16,39] and allows to reduce the hormonal concentration used without affecting the reproductive performance of female rabbits [40]. This method of inducing ovulation in rabbit AI was developed to increase the welfare of rabbit insemination procedures and reduce the concentration of GnRH analogues added to semen extenders.

Against this background, this study aimed to evaluate the antibacterial activity of EDTA, bestatin and chitosan-based nanoparticles added to rabbit semen extender and their effect on the reproductive performance under field conditions, with a view to improving the sustainability of the rabbit production system.

## 2. Materials and Methods

The chemicals used in this study were purchased from Sigma-Aldrich (Merck Life Science S.L.U. Madrid, Spain). Animal housing and the protocols for semen collection and AI were approved by the Animal Care and Use Committee of Centro de Tecnología Animal, Instituto Valenciano de Investigaciones Agrarias. All animals were handled according to the European regulations for the care and use of animals for scientific purposes (European Commission Directive 2010/63/European Union).

### 2.1. Extenders Composition

Four different extenders were tested (Table 1). The solution used as a carrier for the molecules tested was Tris-citric acid-glucose (TCG extender; [41]). Chitosan and alginate were dissolved (0.05%) in the TCG supplemented with EDTA (20 mM) and bestatin (10 mM) according to Casares-Crespo et al. [40]. The nanoparticles were formed spontaneously in the coacervation process, directly after mixing the solutions of chitosan and alginate (4:1) through magnetic stirring (~600 rpm) for 30 min at room temperature.

### 2.2. Experimental Design

#### 2.2.1. Experiment 1: In Vitro Evaluation

##### Animals

Twelve males of New Zealand White origin were kept individually under similar conditions to those described by Viudes de Castro et al. [16].

##### Semen Collection

Semen from males was collected twice a week in three replicates. Strict attention was paid to the hygiene of collection equipment and semen samples were collected into sterile tubes. Semen collection and evaluation were conducted in accordance with Casares Crespo et al. [39]. Finally, all the ejaculates were pooled.

The pool was split into four equal fractions and diluted with the appropriate extender (dilution 1:10; *v*:*v*). The pools used in the experiment presented an average sperm concentration of 385 spermatozoa mL-1. Diluted samples were cooled at 15 °C for three days. Cooled samples were examined for bacterial growth, total motility, percentage of live sperm and membrane status every 24 h for 3 days.

##### Microbiological Analysis

Using Enterobacteriaceae as the sentinel, bacterial growth was evaluated by enumeration. Tenfold dilution series were performed in each extender to 10^−6^, and 1000 µL of each dilution was then plated onto Violet Red Bile Dextrose Agar (VRBD agar, Scharlab^®^, Barcelona, Spain) per duplicate; after homogenisation of the plate, 10 mL of VRBD were added to seal the plate. Plates were incubated for 24–48 h at 37 ± 1 °C. Typical colonies were counted, and the least dilute pair of plates that contained an average of between 30 and 150 colonies was used to calculate the number of bacteria (CFU/mL).

##### Seminal Quality Evaluation

Percentage of total motile sperm was evaluated using a computer-assisted sperm analysis system (ISAS Proiser, Valencia, Spain) as described by Viudes-de-Castro et al. [13]. Briefly, ten microlitres of each sample was placed into a 10 mm deep Makler counting chamber (Sefi Medical Instruments, Haifa, Israel). Sperm motility was assessed at 37 °C by negative phase contrast objective at a magnification of X100 (NIKON E-400 microscope, Izasa Scientific, Barcelona, Spain). Six microscopic fields were captured for each sample. A minimum of 400 sperm were evaluated using the same criteria described by Casares Crespo et al. [39].

Flow cytometric analyses to assess viability (integrity of sperm membrane) and acrosome integrity were performed with a CytoFLEX Flow Cytometer (Beckman Coulter, S.L.U., Barcelona, Spain) equipped with red (638 nm), blue (488 nm) and violet (405 nm) lasers and operated by the CytExpert Software v.2.3 (Beckman Coulter, S.L.U., Barcelona, Spain). The cytometer was calibrated daily using specific calibration beads provided by the manufacturer. Data were collected from 10,000 events. Gating the spermatozoa population after Hoechst 33,342 staining eliminated non-sperm events. Doublets and clumps were further excluded by using a plot of side scatter area and side scatter height followed by a gate of simple events. A compensation overlap was performed before each experiment. A FITC-PNA/PI/Hoechst triple staining method, validated for rabbit semen in our laboratory, was used to determine viability and acrosomal status. To this end, 100 µL of semen at 30 × 10^6^ sperm/mL were stained with 0.5 µL Hoechst 33,342 (0.5 mg/mL) for 20 min at 37 °C without light. Subsequently, 1.5 µL FITC-PNA (1 mg/mL) and 0.5 µL PI (1 mg/mL) were added to the sample and incubated 10 min at 37 °C without light. Then 400 µL of TCG extender [36] were added to obtain a final concentration of 6 × 10^6^ sperm/mL. PI-negative sperm were considered viable. The normal apical ridge (NAR) percentage was calculated as the proportion of acrosome intact sperm.

Membrane functionality analysis was assessed by hypo-osmotic swelling test (HOST). An aliquot of 100 µL of diluted semen was added to 1 mL of warmed 150 mOsm hypo-osmotic swelling solution containing sodium citrate (25 mmol/L) and fructose (75 mmol/L) and incubated for 30 min at 37 °C. Subsequently, 10 µL of each sample were placed on a clean glass slide with a coverslip, and sperm swelling was assessed under phase-contrast microscopy. For each sample, a total of 200 spermatozoa were examined.

#### 2.2.2. Experiment 2: In Vivo Evaluation

Eight hundred and ninety-seven crossbreed females from a commercial farm (Altura, Castellón, Spain) were inseminated using fresh semen from 50 adult males belonging to a paternal rabbit line (Line R, [42]). Animal housing and seminal evaluation were similar to the previous experiment. All the ejaculates were pooled. The pool was split into four aliquots and diluted 1:10 with the four experimental extenders. After diluting the semen in the four experimental extenders, the insemination was initiated immediately. Each female was randomly assigned to one of the four experimental groups and was inseminated with 0.5 mL of semen using standard curved cannulas (24 cm). About 2 h elapsed between the first and the last inseminated female. At birth, pregnancy rate (number of kindlings/number of inseminated does) and prolificacy (total number of kits born) were evaluated.

### 2.3. Statistical Analysis

To analyse the effect of extender on Enterobacteriaceae growth, motility, viability, acrosome integrity and membrane functionality, a general linear model was used. The extender, refrigeration time and their interaction were taken as fixed effects and, in the case of seminal parameters, the corresponding parameter of the pool was introduced as a covariate in the analysis. A chi-square test was used to test differences in pregnancy rate at birth between groups. For the total number of kits born per litter, an ANOVA was performed, including as fixed effect the extender group and pool as covariate. All analyses were performed with the SPSS 26.0 software package (SPSS Inc., Chicago, IL, USA). Values were considered statistically different at *p* < 0.05.

## 3. Results

### 3.1. Experiment 1: In Vitro Evaluation

Results for bacterial growth are shown in Figure 1. The enterobacteria count increased considerably during storage at 72 h (Figure 1) in TCG+AB and TCG-AB groups. At 24 h, a significant increase of bacterial growth was observed in semen extended in TCG-AB (extender without antibiotics) compared to the rest of the groups. Additionally, at this time point, a significant decrease in bacterial growth was observed in the EB group versus the TCG+AB (extender with antibiotics) or QEB groups. At 48 h, it was found that there were no significant differences between EB and QEB, but there was significantly lower bacterial growth than TCG+AB and TCG-AB group. At 72 h, the trend was similar, with no significant differences between EB and QEB groups and showing a significantly lower bacterial growth than TCG+AB and TCG-AB groups. From 24 h, bacterial growth in the TCG+AB group increased over time, being three times higher at 72 h than that observed in groups EB and QEB, which maintained the same number of CFUs over time.

Seminal quality parameters of samples from the experimental extenders are shown in Table 2. There was no interaction between extender and refrigeration time. The presence of EDTA, bestatin and chitosan nanoparticles had no effect on total motility, acrosome integrity, membrane functionality or the viability of spermatozoa. Following 24, 48 and 72 h, quality characteristics were retained in all groups, with no significant differences in motility, acrosome integrity, membrane functionality or the viability of spermatozoa.

### 3.2. Experiment 2: In Vivo Evaluation

Pregnancy rate at birth and the total number of kits born are presented in Table 3. Neither pregnancy rate at birth nor prolificacy were affected by the experimental group, both parameters being similar between groups.

## 4. Discussion

Bacterial contamination is of particular relevance in rabbit AI, where most inseminations are carried out with liquid semen storage at 15 °C [43]. Even though semen collection protocols in the livestock industry are very strict, semen collection is not a sterile process, and the addition of antibiotics to extenders to control contaminating bacterial populations is a routine fact at farm level. The efficacy of different antibiotics added to semen extenders in livestock has been widely demonstrated [44]. However, how quickly bacteria acquire tolerance and/or resistance to antibiotics is essential. Numerous studies show the critical resistance patterns found in semen samples of different species such as boars [45], humans [46] and bovine [47]. However, it can be suggested that the antibiotics currently used in routine practices in livestock, such as AI, may need to be modified to avoid future complications arising from bacterial resistance.

Previous study demonstrated that sperm microbiota diversity is influenced by host genetics [9]. The rabbit semen samples were contaminated with bacteria, especially those that belong to the *Enterobacteriaceae* family [44], ranging from 27.6% for Line V to 50.9% for Line R of semen samples analysed [9]. Bacteria contamination, such as *Enterobacteriaceae* family might adversely affect the quality of semen used for AI [4] and is an important hygiene indicator [5], although the majority of these bacterial strains are not currently considered pathogens [7]. The results of this study prove that the replacement of antibiotics in the current extenders by EDTA and bestatin prevent bacterial growth through 72 h in the rabbit doses. Therefore, the present study results validate the bacteriostatic effect of aminopeptidase inhibitors, such as bestatin and EDTA, and highlight the role of protease inhibitors in the control of seminal bacterial growth. This is in agreement with several studies in which an inhibitory activity of EDTA against Gram-negative bacteria, Gram-positive bacteria (staphylococci) and fungi (*Candida* spp.) was observed [33,34,35]. Likewise, some studies indicated that EDTA, alone or in combination, is an effective antibiofilm agent with a spectrum covering both Gram-positive and Gram-negative bacteria [35,48,49,50,51,52]. In addition, bacterial proteases participate in important metabolic pathways and have key roles in cell viability, stress response and pathogenicity [53]. On the other hand, despite the antimicrobial activity demonstrated by chitosan against several pathogens [54,55,56], in the present study, the presence of chitosan-alginate nanoparticles in the extender did not show a synergistic action with aminopeptidase inhibitors on the microbial growth, with both extenders showing similar results. A possible explanation for this is that the presence of alginate can interfere in the inhibition of bacterial growth by chitosan. As the presence of cationic charge situated in the amino group of chitosan is essential for exhibiting high antimicrobial properties [57], the ionic linkages between functional groups of the oppositely charged chitosan and alginate would result in low availability of unreacted positive amino groups of chitosan when nanoparticles were formed, which reduces the chances of interaction with negatively charged components of microbial cell membranes. Our results suggest that both aminopeptidase inhibitors (EDTA and bestatin), alone or in combination with nanoparticles of chitosan-alginate, maintained total motility, viability, acrosome status and functional integrity of the sperm plasma membrane for at least three days. Moreover, the use of both aminopeptidase inhibitors preserved the fertility and prolificacy under field conditions. Undoubtedly, this result could be used to improve the sustainability of the rabbit production system. In the present study, the classic combination of penicillin and streptomycin contributed to the diluent’s efficacy in controlling the growth of *Enterobacteriaceae* only for up to 24 h. Nevertheless, from this moment on, this antibiotic cocktail cannot prevent the bacterial growth, which was probably due to an increase in tolerance of antibiotics by *Enterobacteriaceae* [58,59]. The common use of antibiotics in extenders is an important concern: apart from being prophylactic and non-therapeutic, and therefore going against the recommendations for prudent use of antimicrobials, they can cause increase antibiotic resistance in the bacteria commonly found in semen [24]. As expected, in the extender without antibiotics (TCG-AB), no bacteriostatic effect was observed throughout the entire cooling period, showing an increasing number of enterobacteria over time, highlighting the need to supplement AI extenders with substances that control bacterial growth. However, there were no effects of enterobacteria contamination on in vitro quality sperm in long-term stored samples. Although the presence of microorganisms in semen may reduce semen quality and fertilising capacity during preservation time, our results indicate that bacterial concentration present in fresh semen (0 to 4 h from semen collection) has no effect on reproductive performance, which is in agreement with Jäkel et al. [60] in pig, who reported similar reproductive performance at 24 h between groups inseminated with semen diluted in extender with or without antibiotics. Extenders and storage temperature are important factors to preserve the fertilising capacity of rabbit semen. As regards the sperm quality variables, all extenders used in the present study preserved the quality of rabbit semen throughout the cooling period. Several studies to evaluate stored rabbit semen have been carried out under different experimental conditions. On the one hand, several authors had observed that motility decreased when semen was stored at 5 °C during 48 h, irrespective of the extender used [61,62,63]. On the other hand, other authors have shown that rabbit semen stored at 15 °C up to 48 h retains fertility capacity [43,64]. However, further studies are needed to verify the reproductive performance of rabbit semen stored for 72 h in extenders supplemented with EDTA and bestatin.

## 5. Conclusions

In conclusion, we demonstrated in this study that the addition of EDTA and bestatin to semen extender exerted a potent bacteriostatic effect over time, effectively inhibiting the growth of Enterobacteriaceae, which suggests that EDTA and bestatin could be used as an alternative to conventional antibiotics in rabbit semen extenders.

## Figures and Tables

**Figure 1 animals-11-01178-f001:**
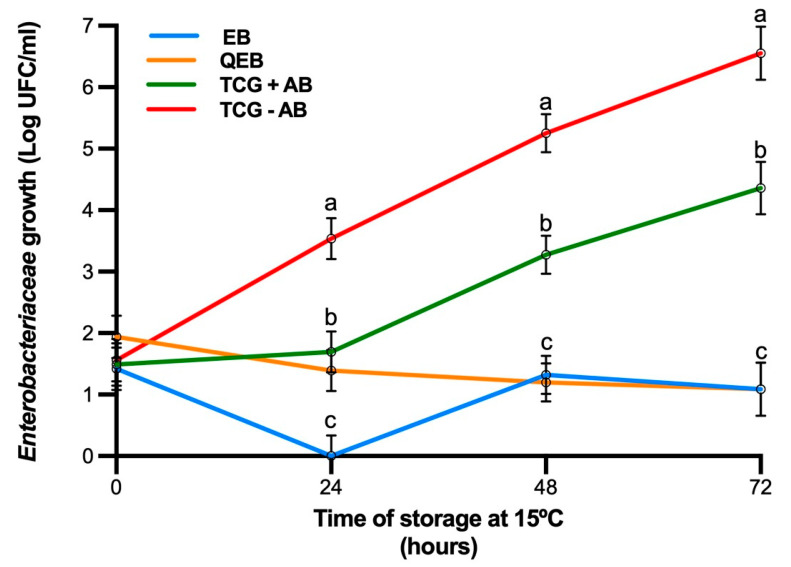
Bacterial growth (Log CFU/mL; mean ± SEM) in sperm samples from the rabbit where there was storage at 15 °C for 72 h in four extenders: EB: TCG supplemented with EDTA (20 mM) and bestatin (10 mM). QEB: TCG supplemented with EDTA (20 mM), bestatin (10 mM) and nanoparticles of chitosan-alginate (0.05%). TCG+AB: TCG extender supplemented with 100 IU/mL penicillin + 100 µg/mL streptomycin. TCG-AB: TCG extender without antibiotics. Different superscripts (a, b and c) indicate differences in values at the same time point (*p* < 0.05).

**Table 1 animals-11-01178-t001:** Semen extenders composition.

Group	Composition
TCG+AB	TCG extender supplemented with 100 IU/mL penicillin + 100 µg/mL streptomycin
EB	TCG supplemented with EDTA (20 mM) and bestatin (10 mM)
QEB	TCG supplemented with EDTA (20 mM), bestatin (10 mM) and nanoparticles of chitosan-alginate (0.05%)
TCG-AB	TCG extender without antibiotics

**Table 2 animals-11-01178-t002:** Sperm quality in stored rabbit spermatozoa in four extenders (means ± standard deviation).

Extender Group	N	Total Mot (%)	HOST (%)	NAR (%)	Viability (%)
TCG+AB	9	73.9 ± 9.98	73.0 ± 9.44	94.2 ± 6.18	75.0 ± 8.38
EB	9	69.0 ± 12.42	73.4 ± 8.41	94.1 ± 5.48	72.0 ± 7.00
QEB	9	71.1 ± 9.48	72.6 ± 6.38	93.7 ± 5.89	72.7 ± 7.43
TCG-AB	9	68.8 ± 11.19	71.7 ± 8.47	94.6 ± 5.79	73.2 ± 6.09
Time					
24 h	12	73.8 ± 11.95	75.1 ± 7.41	96.1 ± 4.96	74.8 ± 6.82
48 h	12	72.0 ± 12.58	71.6 ± 8.78	94.5 ± 6.78	72.6 ± 8.61
72 h	12	66.3 ± 4.43	71.3 ± 7.59	91.8 ± 5.01	72.3 ± 5.76

UTCG+AB: TCG extender supplemented with 100 IU/mL penicillin + 100 µg/mL streptomycin. EB: TCG supplemented with EDTA (20 mM) and bestatin (10 mM). QEB: TCG supplemented with EDTA (20 mM), bestatin (10 mM) and nanoparticles of chitosan-alginate (0.05%). TCG-AB: TCG extender without antibiotics. N: number of seminal pools; Total Mot: total motility; HOST: hypo-osmotic swelling test; NAR: acrosome normality.

**Table 3 animals-11-01178-t003:** Fertility (%) and prolificacy (means ± standard deviation) obtained from insemination of 897 females.

Extender Group	N	Pregnancy Rate (%)	Total Number of Kits Born
TCG+AB	228	90	10.1 ± 3.20
EB	225	88	10.0 ± 3.50
QEB	219	88	10.3 ± 3.17
TCG-AB	225	87	10.2 ± 3.25

TCG+AB: TCG extender supplemented with 100 IU/mL penicillin + 100 µg/mL streptomycin. EB: TCG supplemented with EDTA (20 mM) and bestatin (10 mM). QEB: TCG supplemented with EDTA (20 mM), bestatin (10 mM) and nanoparticles of chitosan-alginate (0.05%). TCG-AB: TCG extender without antibiotics. N: number of inseminated does.

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
