# Peer review of "Antibacterial Activity of Some Molecules Added to Rabbit Semen Extender as Alternative to Antibiotics"

_animals, 2021, doi:10.3390/ani11041178_

Round 1
Reviewer 1 Report
Alternatives to conventional antibiotics in semen extenders are needed, therefore, the present study is important and interesting. However, I miss an explanation of how bacterial resistance arises. Bacteria evolve resistance genes all the time and have done since they first appeared on the planet. The authors do not address whether bacteria could become resistant to these new agents if used instead of conventional antibiotics. The results from the treatment group without antibiotics were similar to the treatment with antibiotics, so why add antibiotics at all? Please expand the discussion to cover these points.
There are several major omissions from the methodology that make interpretation of the data difficult. Why only look for Enterobacteria? Bacteria from the skin and respiratory tract are likely to be present as well. Why culture at 41°? Is this standard for this type of agar? Why was sperm membrane integrity evaluated by HOST when a flow cytometric method would have given a more objective evaluation based on many thousands of spermatozoa? The flow cytometer was available for the other analysis. Chromatin integrity would have given valuable additional information. No details of the CASA analysis are provided. How many sperm were evaluated? What criteria are used to identify a progressively motile sperm? Temperature of evaluation? Chamber type? Prior incubation or not?
More details of the AI trial are needed. How were the various sperm treatments assigned to females? Were the females age- and parity-matched? How were semen samples from different males distributed among the females? Were stored semen samples used or only fresh samples? How long was the semen in contact with the treatments before insemination? Why is male not considered as a factor in the statistical analysis, especially since there were differences in numbers of bacteria found between male lines? A more sophisticated statistical mmodel may be required.
Specific points
Line 104-105: “Strict attention was paid to the collection equipment”: what does this mean?
Line 123-125: repetition; please re-phrase
line 135: is there a reference to this triple staining method?
Reviewer 2 Report
This study investigates whether EDTA and bestatin may be used as a substitute for antibiotics in rabbit semen extender to prevent bacterial growth during chilled storage prior to use in assisted insemination. Of merit is the diversity of response measures examined and the size of the in vivo fertility trial. Suggestions for improvements in the manuscript include further justification of methods and improved reporting of results.
Main comments
Methods:
Extenders used – please provide a justification for the concentration of supplements used. Also, why was each element not tested separately?
Semen collection – more information is required to assess and replicate the methods. For example, Line 108 states that samples were pooled. Was this pooling done across males or just across the two ejaculates per male? How many samples were excluded after subjective assessment? How many samples were used? How were the samples split? Did each sample have an equal volume? Was sperm concentration measured and standardised?
In vivo trial – was chilled semen used? How long between collection and insemination?
Analyses – was sample (Experiment 1) and/or male (Experiment 2) included as a random factor (to account for non-independence of replicates)?
Results:
Figure 1 – no error bars are shown.
Figure 1 – How confident are you that the 24h samples of the EB group had zero bacterial growth? Could this be an error or could have something happened to these samples? It seems strange that this treatment would drop to zero and then rebound to similar levels to the QEB treatment at 48 and 72h.
Table 2 – model outputs of least squares means and s.e. are reported. However, it would be more useful and aid in interpretation if the actual means and s.e. of raw data were shown. In particular, I would be interested in the variance values of each treatment to assess how consistent the response is across samples.
Table 3 – Again, it would be more useful to show raw data than model outputs. If each sample was split 4 ways, why are there differences in the number of inseminated doses among treatments? Was pregnancy success accounted for in total number of kits born?
Were any health measures of the females assessed? Any sign of infection (particularly in the control treatment lacking antibiotics)?
Discussion:
The study concludes that EDTA and bestatin are appropriate substitutes to the use of antibiotics, yet the results show that there was no reduction in sperm quality or fecundity in the TCB-AB treatment. So are bacterial inhibitors required at all (particularly when fresh semen is used)?
What are the downsides of using EDTA and bestatin? The introduction briefly mentions the time cost of other methods used (centrifugation). It would be good to also include details on potential barriers to the use of the supplements tested (e.g. cost, availability).
Minor comments
Line 20 – please define EDTA before using the abbreviation.
Line 62 – Comma instead of full stop
Table 1 – 100 IU/mL (not UI?)
Line 119 – please define how ‘typical’ colonies were selected
Line 292 – add ethics statement
Round 2
Reviewer 1 Report
The authors have addressed my comments satisfactorily. I have no further comments to add.